# Sinkhole Attack Defense Strategy Integrating SPA and Jaya Algorithms in Wireless Sensor Networks

**DOI:** 10.3390/s23249709

**Published:** 2023-12-08

**Authors:** Zhijun Teng, Mingzhe Li, Libo Yu, Jinliang Gu, Meng Li

**Affiliations:** 1Modern Power System Simulation Control and Green Energy New Technology Key Laboratory of the Ministry of Education, School of Electrical Engineering, Northeast Electric Power University, Jilin 132012, China; tengzhijun@163.com; 2School of Electrical Engineering, Northeast Electric Power University, Jilin 132012, China; 13180963381@163.com (L.Y.); gujinliang1997@163.com (J.G.); 15765594470@163.com (M.L.)

**Keywords:** wireless sensor network, set pair analysis, Jaya algorithm, sinkhole attack, defense

## Abstract

A sinkhole attack is characterized by low difficulty to launch, high destructive power, and difficulty to detect and defend. It is a common attack mode for wireless sensor networks. This paper proposes a sinkhole attack detection and defense strategy integrating SPA and Jaya algorithms in wireless sensor networks (WSNs). Then, combined with the SPA trust model, the trust values of suspicious nodes were calculated, and the attack nodes were detected. The Jaya algorithm was adopted to avoid the attacked area so that nodes can find the route to communicate with the real Sink, and attack nodes are isolated in the network to improve the capabilities of network directional defense. The simulation results show that the improved detection algorithm can effectively detect malicious nodes in the network, and the defense strategy implemented in the attacked area can improve the packet delivery rate, reduce network delay and energy consumption, and improve the security and reliability of wireless sensor networks.

## 1. Introduction

A wireless sensor network (WSN) is a network composed of many sensing nodes [1]. Its nodes have communication and computing capabilities and can cooperate in real-time monitoring, sensing, collecting information about various environments or monitoring objects in the network distribution area, and processing this information to obtain detailed and accurate information, which can be transmitted to users who need this information. It can be widely used in national defense and military affairs, national security, environmental monitoring, traffic management, health care, manufacturing, anti-terrorism, disaster relief, and other fields [2]. However, due to limited sensor node resources and often being deployed in unsupervised areas, it is vulnerable to various forms of attacks [3,4,5,6]. A sinkhole attack (Sinkhole) provides a bridge to a wide range of internal attacks, attracting almost all the surrounding data streams to pass through the node and performing malicious operations on the received data, which is highly destructive [7,8]. As a result, network security is one of the most challenging issues in WSNs.

The literature [9] proposed a detection algorithm DFFR based on data traffic, which used data mining techniques to analyze the packet traffic and forwarding rate in the network to detect sinkhole attacks, taking advantage of the significantly larger data traffic within the vortex and the lower packet forwarding rate of the malicious nodes. The literature [10] proposed a PRDSA scheme for defending against sinkhole attacks based on probing routes, which combined the routing mechanisms of far-Sink reverse routing, equal-hop routing, and minimum-hop routing to be able to circumvent sinkhole attacks and found a secure route to the real Sink. The literature [11] proposed a HCODESSA detection scheme for sinkhole attacks based on hop counting, where the base station received all the hop counts from neighboring nodes and sorted the hop counts in ascending order, which caused the node hop count sorting position to change after being attacked, and identified Sinkhole nodes according to the position change. Another paper [12] proposed a hybrid intrusion detection system with PC and MK-Means machine learning techniques, where data pre-processing was performed in PCS to reduce the amount of collected training data features, and the pre-processed features were sent to the MK-Means algorithm for data training and classification enhancement, capable of capturing Sinkhole and Blackhole nodes. The authors of [13] proposed the trust-based RPL secure routing protocol RF-Trust, which used RF algorithms to identify the trustworthiness of nodes in this model and subjective logic to avoid biased or incorrect recommendations. Trust metrics such as the delivery ratio, delay, energy consumption, and honesty were used to detect sinkhole attacks in IoT networks. Other authors [14] proposed a lightweight and secure approach based on a threshold-sensitive high-performance sensor network protocol and watermarking techniques, using watermarking to protect sensed data during transmission, homomorphic encryption processing for internal communication, and the use of network keys for Sinkhole node detection. The authors of [15] proposed a detection method based on contaminated boundaries. The proposed scheme relied on the assumption that contaminated areas and contaminated boundaries exist and used the sequence number as the target feature to calculate the difference value between the sequence number on the contaminated boundary node and the sequence number of its neighbors to identify the malicious behavior of the Sinkhole node. Next, ref. [16] proposed RMHSD, a Sinkhole detection algorithm based on minimum hop selection random paths, which measured the frequency of each node by establishing M optimal hop routes from each node to the Sink node. Dynamic programming was used to build a database, calculate the hop difference between suspect nodes and their neighbors, and compare it with the threshold value to identify malicious nodes. In contrast, ref. [17] used a centralized geostatistical fragile survival model approach to obtain the residual energy of sampled nodes and their geostatistical data to detect suspicious areas in the network, and a distributed monitoring approach using fully trusted surveillance nodes monitoring local information was used to explore each neighborhood in the network to detect malicious behavior.

Some of the existing research is not effective against sinkhole attacks due to the great danger of sinkhole attacks and the difficulty in detecting and defending against them. As a result, the following issues still require improvement:(1)In the existing research, there are several methods that can detect sinkhole attacks. However, some studies can only detect the presence of a Sinkhole in a network but cannot determine the location of the Sinkhole.(2)It is possible to bypass the scope of a sinkhole attack to reach the Sink using a multi-path forwarding approach, but using multiple routes at the same time can seriously damage the network’s lifespan. Even when multiple routes are used, it is difficult to ensure that one route will bypass the sinkhole attack.(3)When a Sinkhole does not perform any attacks on other nodes, neighboring nodes cannot observe the Sinkhole’s illegal behavior. Even if abnormal behavior is observed, warning messages from nodes within the attack range are sent to the Sinkhole, and therefore, the damage caused to the network by the sinkhole attack cannot be eliminated in time. As a result, the network still does not have access to valid warning messages.(4)A node in range of an attack cannot report to the Sink whether it is under attack and requires additional hardware or other policies in the network to report the presence of a Sinkhole in the network. This approach will result in increased network costs and reduced network performance.

This paper proposes a sinkhole attack detection and defensive strategy integrating the SPA and Jaya algorithm in WSNs (SJ-SHDDS) with the innovation of using a segmented detection method. Based on the attack characteristics of sinkhole attacks, we first analyzed the number of node hops in the network and then combined the SPA trust model to calculate the trust value of suspicious nodes and identify the attacking nodes. Finally, to avoid retransmitting packets to the attacking node, the Jaya algorithm was used to avoid the attacked area, find the optimal path to communicate with the real Sink node, and isolate the attacking node in the whole network.

## 2. Analysis of Typical Cyber Attacks

### 2.1. Network Model

This paper uses a typical wireless sensor network model [18] where a Sink node is deployed in the network, keeping the location information of all nodes, with unlimited energy and high transmitting power to communicate directly with the nodes. All sensor nodes in the network are randomly distributed with density ρ and are homogeneous, where *r* is the communication radius. *E_S_* is the initial energy. The storage capacity and the computational power are the same. Each node is assigned a unique ID and a fixed location that cannot be moved after deployment, and each node can communicate with its neighbors within the communication radius and obtain the neighbor’s ID and the number of hops to the Sink. The sensor network is of the event-monitoring type. After an event occurs, the node that detects the event will generate a packet, and the sensor node collects the information and transmits it to the Sink node via a shortest path routing algorithm.

The distribution density of sensor nodes is as follows:(1)$$\rho =\frac{N_{node}}{S}$$
where *N_node_* is the number of nodes and *S* is the approximate uniformly distributed area of each node.

The energy consumption of the sensor nodes is modeled as follows:(2)$$E_{r}\left(l,d\right)=E_{elec}\times l$$
(3)$$E_{s}\left(l,d\right)=E_{elec}\times l+E_{amp}\times ld^{2}$$
(4)$$E_{C}\left(l,d\right)=2\cdot E_{elec}\times l+E_{amp}\times ld^{2}$$
where *E_r_* (*l*, *d*) and *E_s_* (*l*, *d*) are the energy consumed by the node when sending and receiving data, respectively. *E_C_* (*l*, *d*) is the total energy consumption of the node when performing data forwarding. *l* is the number of packet bits sent and received. *d* is the distance between node s and node *f*. *E_elec_* is the energy per unit bit consumed by node *j* when transmitting data, and *E_amp_* is the energy consumed by the power amplifier during transmission. If the initial energy of the network node is *E_B_*, the remaining energy of the node is
(5)$$E_{residual}\;\left(l,d\right)=E_{B}-E_{C}\left(l,d\right)$$

### 2.2. Sinkhole Attack Model

A model of a wireless sensor network under a sinkhole attack [19] is shown in Figure 1, where SH is the malicious node and Sink is the aggregation node. Figure 1a shows the data flow in the network without any attack, and Figure 1b shows the data flow in the network with a sinkhole attack.

The sinkhole attack in Figure 1b attracts a large amount of data by broadcasting a false message, creating the illusion of a channel with high quality to the base station. Nodes that receive this message are lured to send their own data to the attacking node, and the lured sensor nodes may continue to spread this false message within their own communication range, again attracting some normal nodes to send packets to the attacker, thus forming a metaphorical Sinkhole around the attacker. A malicious node can arbitrarily organize or tamper with the packets sent by the attacking area nodes, thus posing a significant threat to WSN security.

## 3. SJ-SHDDS Algorithms

The SJ-SHDDS algorithms proposed in this paper can be divided into three phases—suspiciousness detection phase, malicious identification phase, and defense against attack phase.

### 3.1. Suspiciousness Detection

In WSNs, routing from sensor nodes to the Sink is usually based on a hop-by-hop routing mechanism. When data need to be routed to the Sink, the sensor node chooses the neighboring node with the smallest number of hops to the Sink to forward the data. However, SH nodes usually set their hop count to the Sink to be less than their actual hop count and broadcast a false hop count to all neighboring nodes to detect anomalies based on hop count changes.

Definition 1—Boundary nodes: Using Equations (6) and (7) to calculate the degree of suspiciousness between the average hop count *Hop_average_* and the lowest hop count *Hop_min_*, if the *Suspiciousness* degree is greater than the threshold *Suspicious_th_*, the node is defined as a boundary node—that is, the node is a neighboring node contaminated by the attacking node.
(6)$$Hop_{average}=\frac{Hop_{1}+Hop_{2}+\ldots +Hop_{n}}{n}$$
(7)$$Suspicious\;\;\left(\%\right)=(1-\frac{Hop_{min}}{Hop_{average}})\times 100\%$$
where *n* is the number of neighboring nodes. *Hop_n_* is the hop count of the nth node. And since the hop count of Sink is 0, the suspiciousness of Sink’s neighboring nodes is 1.

As shown in Figure 2, red nodes are SH nodes, grey nodes are boundary nodes, and the rest of the nodes are normal nodes. Taking node 5 as an example, the degree of suspiciousness values of the nodes are analyzed, as shown in Table 1.

Definition 2—Suspicious node: There is no significant difference between the hop count from a node to a Sink and the hop count from a neighboring node to a Sink. If the neighboring node has an abnormally low hop count, the neighboring node is determined to be suspicious. Thus, the node with the lowest hop count among the neighboring nodes of the boundary node is the suspicious node.

### 3.2. Building a Trust Evaluation Model

This phase is the malicious identification phase, in which the trust value of a suspicious node in the path is calculated using a boundary node at a fixed time interval to determine the state of the node. The trust degree between nodes is subjective and fuzzy, and the main methods of trust evaluation are hierarchical analysis [20] and fuzzy comprehensive judgment [21]. In this paper, we use set pair analysis (SPA) to use the change of the operation state of nodes to calculate the comprehensive trust value of nodes.

#### 3.2.1. SPA

SPA is a mathematical theory that deals with the interaction of certainty and uncertainty [22] and is able to analyze the effects of uncertainty elements such as randomness, non-linearity, and ambiguity, which is suitable for evaluating trust between nodes.

The expression for SPA connectivity is
(8)$$\mu \left(A,B\right)=\frac{S}{N}+\frac{F}{N}i+\frac{P}{N}j=a+bi+cj$$
where *μ*(*A*, *B*) is the degree of association of the set pair H(A, B). N is the total number of elements in the set, N = S + P + F. S is the number of elements in the same state (same characteristics) in both sets; F is the number of elements in the different state (different characteristics) in both sets; P is the number of elements in the opposite state (opposite characteristics) in both sets; a is the degree of similarity; b is the degree of difference; c is the degree of opposition, and a,b,c ∈ [0, 1], a + b + c = 1. i is the coefficient of the degree of difference, i ∈ [−1, 1]; j is the coefficient of the degree of opposition, taking a constant value of −1.

Determining the similarity of two sets is ultimately achieved by calculating the magnitude of the degree of connectivity. When the weights of each influencing factor or evaluation indicator are considered, then the degree of connectivity μ for the weights of the same, opposing, and differential characteristics is
(9)$$\mu \left(A,B\right)=\sum_{k=1}^{S}\omega_{k}+\sum_{k=S+1}^{S+F}\omega_{k}i+\sum_{k=S+F+1}^{N}\omega_{k}j$$
where $\sum_{k=1}^{N}\omega_{k}=1$.

#### 3.2.2. Direct Trust

This paper uses the SPA method to calculate the trust value of nodes and introduces a reputation maintenance function to adaptively reduce the impact of the number of interactions between nodes in the early stage. Also, since the network nodes may be uncooperative influences brought about by non-intrusive factors—that is, abnormal behavior of the nodes due to the network’s own failures—an abnormal weakening factor is introduced to reduce the false detection rate of the network. The direct trust *DT_ij_* obtained from the direct interaction behavior of the nodes is expressed as follows:(10)$$DT_{sf}(t)=\frac{\kappa a_{sf}+\Delta a_{sf}}{G}+\rho (\frac{\kappa b_{sf}+\Delta b_{sf}}{G}i+\frac{\kappa c_{sf}+\Delta c_{sf}}{G}j)$$
(11)$$\kappa =\frac{\theta }{a_{sf}+b_{sf}+c_{sf}}$$
(12)$$\rho =\frac{N_{int}}{N_{det}}$$
where *a_sf_*, *b_sf_*, and *c_sf_* are the number of historical successful, uncertain, and failed communications between node *s* and node *f*, respectively, ∆*a_sf_*, ∆*b_sf_*, and ∆*c_sf_* are the number of successful, uncertain, and failed communications between nodes in ∆*t* time, and *G* is the total number of node interactions. *κ* is the reputation maintenance function, which maintains the impact of the current node behavior on reputation and reduces the impact of historical behavior. *θ* is a fixed maintenance value to set the range of the maintenance function action. *ρ* ∈ [0, 1] is the suspicious weakening factor. *N_int_* denotes the number of node anomalous communications due to intrusion factors, and *N_det_* denotes the total number of anomalous communications in the network.

#### 3.2.3. Indirect Trust

Three evaluation factors are selected to weigh the indirect trust value of the nodes. The set of evaluation factors *Q* = {*q*_1_, *q*_2_, *q*_3_}, where *q_m_* (*m* = 1, 2, 3) denotes the data delivery rate, node residual energy, and processing delay, respectively. The corresponding weight sets *w* = (*w*_1_*, w*_2_, *w*_3_). *w_m_* (*m* = 1, 2, 3) denotes the weights of each evaluation factor *q_m_*, respectively, and the importance of each factor is calculated by referring to the three-scale method [23] to establish a judgment scale table, which yields *w*_1_ = 0.6334,*w*_2_ = 0.2605,*w*_3_ = 0.1061. The set of trust judgment indicators *E* = {*e*_1_, *e*_2_, *e*_3_}, *e_m_* (*m =* 1, 2, 3) denote untrustworthy, uncertain, and high trustworthiness, respectively. The factors are normalized to the interval [0, 1] and used as input variables, respectively, and the homogeneous inverse vector matrix *R* of each factor can be obtained according to the affiliation function.

Data-normalization process: The evaluation factor set obtains different values for each, and the data need to be normalized by the deviation, a linear transformation of the original data that maps the results to between 0 and 1. The normalization formula is
(13)$$x_{n}=\left\{\begin{array}{cc} 0, & x<x_{\min}\\ \frac{x-x_{\min}}{x_{\max}-x_{\min}}, & x_{\min}\leq x<x_{\max}\\ 1, & x\geq x_{\max} \end{array}\right.$$
where *x_n_* is the normalized deviation value of each evaluation factor index, *x*_max_ and *x*_min_ are the limit values of each corresponding evaluation factor index interval, and each operation state affiliation function is shown in Equations (14)–(16).

(a) The affiliation function when the state is not credible is
(14)$$c_{m}=\left\{\begin{array}{cc} 1, & x_{n}\leq 0.2\\ \frac{1}{2}-\frac{1}{2}\sin\frac{\pi }{0.2}\left(x_{n}-0.3\right), & 0.2<x_{n}<0.4\\ 0, & x_{n}\geq 0.4 \end{array}\right.$$

(b) The affiliation function when the state is uncertain is
(15)$$b_{m}=\left\{\begin{array}{cc} 0, & x_{n}\leq 0.2\;\mathrm{or}\;x_{n}\geq 0.8\\ \frac{1}{2}+\frac{1}{2}\sin\frac{\pi }{0.2}(x_{n}-0.3), & 0.2<x_{n}<0.4\\ 1, & 0.4\leq x_{n}\leq 0.6\\ \frac{1}{2}-\frac{1}{2}\sin\frac{\pi }{0.2}(x_{n}-0.7), & 0.6<x_{n}<0.8 \end{array}\right.$$

(c) The affiliation function when the state is a credible state is
(16)$$a_{m}=\left\{\begin{array}{cc} 0, & x_{n}\leq 0.6\\ \frac{1}{2}+\frac{1}{2}\sin\frac{\pi }{0.2}(x_{n}-0.7), & 0.6<x_{n}<0.8\\ 1, & 0.8\leq x_{n}\leq 1 \end{array}\right.$$

The homogeneous inverse evaluation matrix *R* of the indirect trust *IT_sf(t)_* is constructed by calculating the affiliation functions of the respective ranks according to Equations (14)–(16).

By forming *Q* and *E* into a set of set pairs H(*Q*, *E*) according to Equation (8), the properties of the sets *Q* and *E* are treated as a system, and the identities, differences, and opposites of the sets *Q* and *E* in the set pairs are analyzed. To enable a more intuitive and convenient calculation of the degree of connectivity *μ* when considering weights, the expression for the degree of connectivity is based on the relationship between congruence, dissimilarity, and inverse connectivity as
(17)$$\begin{array}{ll} \mu (Q,E)\; & =WRI\;\;\\ & =\left(\omega_{1},\omega_{2},\omega_{3}\right)\left(\begin{array}{c} \begin{array}{ccc} a_{1} & b_{1} & c_{1} \end{array}\\ \begin{array}{ccc} a_{2} & b_{2} & c_{2} \end{array}\\ \begin{array}{ccc} a_{3} & b_{3} & b_{3} \end{array} \end{array}\right)\;\;\left(\begin{array}{c} 1\\ i\\ j \end{array}\right)\\ & =\left(a_{1}\omega_{1}+a_{2}\omega_{2}+a_{3}\omega_{3}\right)\\ & +\left(b_{1}\omega_{1}+b_{2}\omega_{2}+b_{3}\omega_{3}\right)\;i\\ & +\left(c_{1}\omega_{1}+c_{2}\omega_{2}+c_{3}\omega_{3}\right)j \end{array}$$
where *μ*(*Q*, *E*) is the connectivity of the set pair H(*Q*, *E*), *W* is the vector matrix of weight coefficients, and *I* is the matrix of homogeneous inverse coefficients.

Using the evaluation index data as input and the value of the connectivity *μ* calculated by the process of SPA as output, the range of values of the connectivity [−1, 1] was divided into three different intervals, A = [−1, −0.333], B = [−0.333, 0.333], and C = [0.333, 1], according to the principle of mean score [24], with A, B, and C representing the trust levels of untrustworthiness, uncertainty, and trustworthiness, as shown in Table 2. When the level of indirect trust is uncertain, the trend of level transformation will be considered, and a comparison between the pessimistic and optimistic potential will determine which trend the indirect trust tends to follow before determining the fixed value of indirect trust using the following formula.

Pessimistic potential: a trend where uncertainty translates into untrustworthiness.
(18)$$negative=\frac{c_{1}\omega_{1}+c_{2}\omega_{2}+c_{3}\omega_{3}}{\left(b_{1}\omega_{1}+b_{2}\omega_{2}+b_{3}\omega_{3}\right)+\left(c_{1}\omega_{1}+c_{2}\omega_{2}+c_{3}\omega_{3}\right)}$$

Optimistic potential: a trend where uncertainty translates into trustworthiness.
(19)$$positive=\frac{a_{1}\omega_{1}+a_{2}\omega_{2}+a_{3}\omega_{3}}{\left(a_{1}\omega_{1}+a_{2}\omega_{2}+a_{3}\omega_{3}\right)+\left(b_{1}\omega_{1}+b_{2}\omega_{2}+b_{3}\omega_{3}\right)}$$

#### 3.2.4. Comprehensive Trust

After the calculation of indirect trust is completed, the direct trust and indirect trust are synthesized to obtain the combined trust value of the evaluated node, and the formula for the combined trust value of the node is as follows:(20)$$CT_{sf}\left(t\right)=\gamma \;DT_{sf}\left(t\right)+\left(1-\gamma \right)\;\;IT_{sf}\left(t\right)$$
where γ = 0.5 is the trust weight, and the comprehensive trust value takes the value in [0, 1]; the higher the trust value, the higher the reliability of the node. When a node is attacked, it causes the comprehensive trust value of the node to drop sharply and will be detected when it is less than the trust threshold K.

### 3.3. Jaya-Based Defense Attack Model

In many previous studies, there was an assumption that once anomalous information was detected, the detection results could be transmitted to the real Sink. However, in an actual sinkhole attack, if the original route was used, the detection results would still be transmitted to the fake Sink. This will block communication between nodes within the attack range of the Sinkhole and nodes within the unaffected area, forming a so-called “isolated island” that can only detect sinkhole attacks and cannot provide a method for reporting detection information to the Sink. This is not very applicable in practical applications. Therefore, this paper combines the strategy of updating the solution of the Jaya algorithm to find an optimal path that can avoid the attack region and reach the real Sink.

#### 3.3.1. Strategy for Updating the Solution of the Jaya Algorithm

The Jaya algorithm has only one stage and has the advantages of running parameter-free, solving fast, and not easily falling into local optima [25]. The algorithm strives to win by reaching the optimal solution, and the basic idea of the Jaya algorithm is to converge to the optimal solution and stay away from the worst solution. The idea of this paper is to find the shortest path to the Sink, hiding boundary nodes to avoid re-entering the malicious-node-contaminated area to affect packet transmission. Therefore, the improved Jaya algorithm in this paper updates the path node adaptation as follows:(21)$$\begin{array}{rr} Hop_{Next,iter}\; & =Hop_{Node,iter}\\ & +Suspicious_{best,iter}(Hop_{best,iter}-Hop_{Node,iter})\\ & -Suspicious_{worst,iter}(Hop_{worst,iter}-Hop_{Node,iter}) \end{array}$$
where *Hop_Node_*_,*iter*_ is the original node hop count. *iter* is the current iteration count. *Hop_Node_*_,1_ is the hop count of the boundary node where the malicious node is found. *Suspicious_best_*_,*iter*_ and *Suspicious_worst_*_,*iter*_ are the suspiciousness values of the nodes with the smallest hop count and the greatest suspiciousness, respectively, taking values between [0, 1]. *Hop_best_*_,*iter*_ and *Hop_worst_*_,*iter*_ are the hop values of the nodes with the smallest hop count and the greatest suspiciousness, respectively. *Hop_Next_*_,*iter*_ is the fitness of the updated path node. If the generated new fitness is better than the original node, the original node is replaced with the node found to have the smallest hop count; otherwise, it is not replaced. Then, Jaya is calculated for the second fewest hop count until the optimal path is transmitted to the Sink. This paper does not take the node of the previous hop of the path into account in the calculation.

#### 3.3.2. Defensive Strategies to Circumvent SH Nodes

To find the optimal path under the Sinkhole attack as shown in Figure 3, red nodes are SH nodes, grey nodes are boundary nodes, yellow nodes are the nodes in the optimal path, and the rest of the nodes are normal nodes. Figure 3a shows the boundary nodes affected by the SH node, while Figure 3b shows finding the optimal path while avoiding the sinkhole attack.

The pseudocode of SJ-SHDDS algorithm is shown in Algorithm 1.
**Algorithm 1:** SJ-SHDDS Algorithm Description1. begin2. while true do
 3.     for *i*∈true % *i represents the neighboring node of the boundary node*
 4.     query its neighboring nodes5.      for *p*,*q*∈Γ(*i*) % *p, q represents the neighbor node of i*6.     if *p*=max[*Suspicious*] %*Suspicious* for suspicion7.      max[*Suspicious*]=*Suspicious_worst_* 8.      *p*[*hop*]=*Hop_worst_* %*p*[*hop*] *is the number of hops of p*9.     endif10.     remove p from neighboring nodes11.    if *q*=min[*hop*] %*hop is the number of hops*12.     min[*hop*]=*Hop_best_* 13.        *q*[*Suspicious*]=*Suspicious_best_* %*q*[*Suspicious*] is the degree of suspicion of q14.    endif15.     endfor16.   for *j*=1:iter17.   $\begin{array}{l} Hop_{Next,iter}\\ =Hop_{Node,iter}\\ \;+Suspicious_{best,iter}(Hop_{best,iter}-Hop_{Node,iter})\\ \;-Suspicious_{worst,iter}(Hop_{worst,iter}-Hop_{Node,iter}) \end{array}$
18.   endfor19.   if *Hop_Next_*_,*iter*_≥*Hop_Node_*_,*iter*_20.     Use n node as the next hop node21.   else pick the second small hop node22.      reture *Hop_Next_*_,*iter*_23.   endif24.  endfor25. endwhile

The SJ-SHDDS algorithm process is shown in Figure 4.

## 4. Simulation Results and Performance Analysis

In this paper, MATLAB 2020b is used to build the simulation environment. A square with 100 m as the side length is used as the monitoring area. A total of 100 sensor nodes are randomly placed in this area. The communication range of the nodes is 15 m; malicious nodes drop packets with a probability of 0.1 to 0.5. The evaluation of node trust value is conducted after each time period of operation. Each time period is divided into t time slot. Each time slot is recorded once for forwarding rate, and a certain percentage of malicious nodes are randomly selected for performance analysis. The specific simulation parameters are shown in Table 3.

In this paper, the algorithm performance is judged using the detection rate *DR*, false detection rate *FPNR*, packet delivery rate *PDR,* and end-to-end delay *EED*, as follows:(22)$$PDR=\frac{\mathrm{Packets}\;\mathrm{received}\;\mathrm{by}\;\mathrm{Sin}k}{\mathrm{Packets}\;\mathrm{sent}\;\mathrm{by}\;\mathrm{the}\;\mathrm{source}\;\mathrm{node}}$$
(23)$$\begin{array}{l} FPNR=\frac{\mathrm{Normal}\;\mathrm{node}\;\mathrm{misjudged}}{\mathrm{Total}\;\mathrm{number}\;\mathrm{of}\;\mathrm{nodes}}+\\ \frac{\mathrm{Malicious}\;\mathrm{nodes}\;\mathrm{missed}\;\mathrm{detection}}{\mathrm{Total}\;\mathrm{number}\;\mathrm{of}\;\mathrm{nodes}} \end{array}$$
(24)$$PDR=\frac{\mathrm{Packets}\;\mathrm{received}\;\mathrm{by}\;\mathrm{Sin}k}{\mathrm{Data}\;\mathrm{packets}\;\mathrm{sent}\;\mathrm{by}\;\mathrm{the}\;\mathrm{source}\;\mathrm{node}}$$
(25)$$EED=\frac{\sum_{l=1}^{L}{\mathrm{time}\;\mathrm{delay}}_{l}}{L}$$
where *L* is the total number of nodes in the path.

### 4.1. Selection of Suspiciousness Threshold

As the number of SH nodes M increases, more boundary nodes are affected, as shown in Figure 5. It can be seen that the detection of boundary nodes is missed due to the larger threshold value, resulting in a gradual decrease in the detection rate of boundary nodes with the increase in the threshold value of suspiciousness. As can be seen in Figure 6, the false detection rate of boundary nodes increases after decreasing first, which is due to the fact that when the suspiciousness threshold is less than 0.35, the nodes appearing as normal nodes are mistakenly detected as boundary nodes. When the suspiciousness threshold is greater than 0.35, the boundary nodes are missed as normal nodes. Combining the detection rate and false detection rate of boundary nodes, the suspiciousness threshold *Suspicious_th_* of 0.35 is chosen in this paper.

### 4.2. Selection of Trust Threshold

In Figure 7 and Figure 8, it can be seen that the detection rate of SH nodes gradually increases with the increase in the trust threshold, and the detection rate reaches the highest and remains stable after the trust threshold is greater than 0.45 in both cases. The false detection rate decreases and then increases as the trust threshold increases due to missed detection at trust thresholds less than 0.45 and false detection at trust thresholds greater than 0.45, resulting in inadequate detection of SH nodes. In summary, when the trust threshold is at 0.45, the detection rate is the highest and the false detection rate is the lowest, and SH nodes can be effectively detected, so 0.45 is chosen as the trust threshold in this paper.

### 4.3. Comparative Performance Analysis of Different Algorithms

To verify the effectiveness and security of the algorithm in this paper, an experimental comparative analysis is performed with algorithms such as HCODESSA [11], ICZSHD [15], and RMHSD [16].

In Figure 9 and Figure 10, it can be seen that with the increase in the number of SH nodes, the detection rate of all four methods tends to decrease, and the false detection rate tends to increase, but the detection rate and false detection rate of this paper’s method are better than the other three methods. This is due to the fact that the method in this paper uses a segmented detection method to detect boundary nodes using anomalies and then combines the SPA method to calculate the comprehensive trust value of suspicious nodes, which can distinguish normal nodes from SH nodes and improve the accuracy of detection. Other algorithms directly judge the state of nodes through differences, such as the number of sinkhole attack hops and sequence numbers, without comprehensively considering changes in indicators such as the forwarding rate and energy consumption of nodes in the attack state. The number of neighboring nodes affected by SH nodes increases with the number of SH nodes, which can easily be incorrectly detected as SH nodes, increasing the false detection rate.

As can be seen in Figure 11, the sinkhole attack leads to a decreasing trend and gradual deterioration of the packet delivery rate. The other three algorithms still transmit packets on the original path according to the attraction of SH nodes after the discovery of SH nodes and are unable to transmit packets to the real Sink, resulting in a decreasing packet delivery rate as the number of SH nodes increases. The defense algorithm proposed in this paper uses the Jaya algorithm to circumvent the SH nodes and boundary nodes in the attack area, which can effectively defend against the sinkhole attack of compromised nodes inside and guarantee the normal delivery of packets.

The sinkhole attack leads to an increase in end-to-end delay. The three compared algorithms in Figure 12 have no defense measures, and packet loss is severe as the number of malicious nodes increases, which will lead to nodes needing to re-establish connections and transmit packets, increasing the end-to-end delay. In contrast, the SJ-SHDDS algorithm in this article can effectively defend against sinkhole attacks, and the selection of the optimal path slows down the end-to-end delay growth rate caused by data retransmission.

As can be seen in Figure 13, the mean remaining energy of the network tends to decrease with the increase in the number of packet transmissions. RMHSD and HCODESSA use a multi-path forwarding method with excessive node energy consumption, and the mean remaining energy decreases more rapidly; IDZSHD, with the increase in malicious nodes, leads to energy voids in the contaminated area, both accelerating the energy consumption of network nodes. In contrast, the optimized SJ-SHDDS attack detection and defense mechanism can accurately complete the detection and elimination of SH nodes, effectively reducing the energy consumption of network nodes.

## 5. Conclusions

With the widespread use of wireless sensor networks, security issues have become increasingly important. In this paper, we propose a sinkhole attack detection and defensive strategy for WSNs that incorporates SPA and Jaya algorithms, which considers the abnormal changes in hop count, packet forwarding rate, energy consumption, and delay caused by sinkhole attacks to identify SH nodes and, with the basic idea that Jaya algorithms converge to the optimal solution and stay away from the worst solution, find the optimal path to reach the real Sink and circumvent the attacked area, including the boundary nodes. Simulation results show that the SJ-SHDDS algorithm is able to effectively detect SH nodes and defend against sinkhole attacks on compromised internal nodes compared to other algorithms, guaranteeing the secure and reliable operation of wireless sensor networks. There are still many problems to be studied and solved in the detection of sinkhole attacks in wireless sensor networks, and the next phase will focus on improving the trust evaluation model for the sinkhole attack method of cluster-based wireless sensor networks to improve the security of the network.

## Figures and Tables

**Figure 1 sensors-23-09709-f001:**
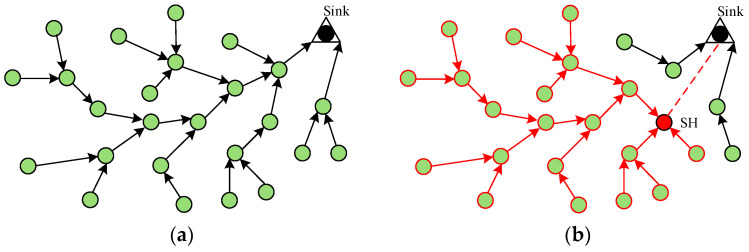
(**a**) No attack on the network; (**b**) sinkhole attack. The black arrow refers to the communication path that is not affected by the SH node, the red arrow refers to the communication path that is affected by the SH node, the green node represents the normal node, the red node represents the SH node, and the black node represents the sink node.

**Figure 2 sensors-23-09709-f002:**
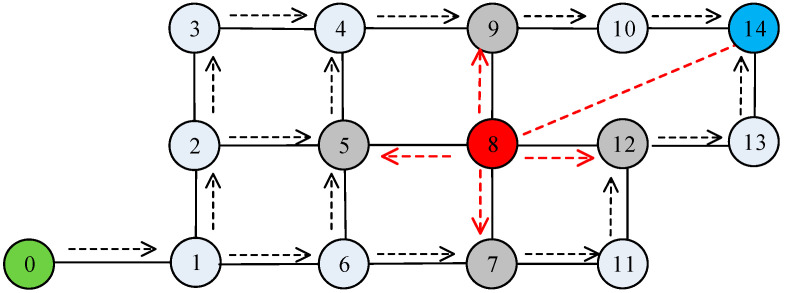
Network topology. The red dotted line refers to the boundary nodes affected by the SH node.

**Figure 3 sensors-23-09709-f003:**
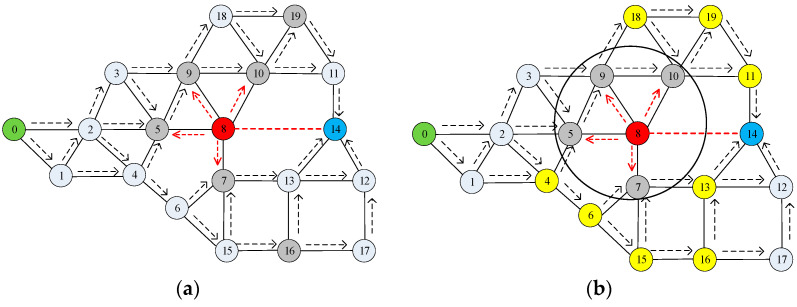
(**a**) A boundary node in network; (**b**) The optimal path to circumvent the attacked area.

**Figure 4 sensors-23-09709-f004:**
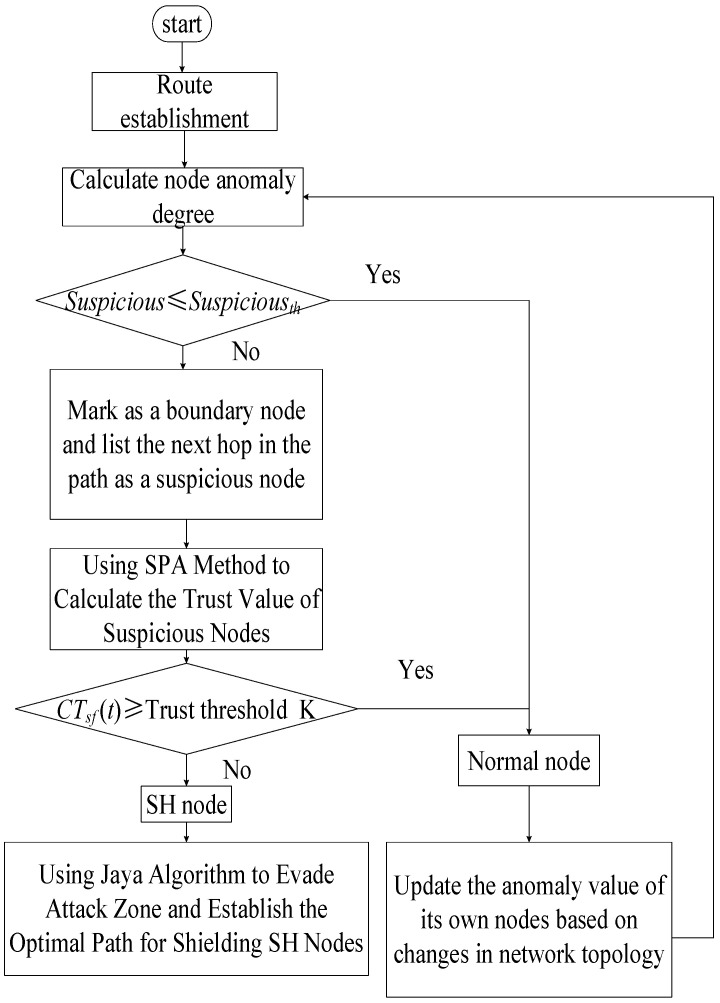
SJ-SHDDS algorithm flow chart.

**Figure 5 sensors-23-09709-f005:**
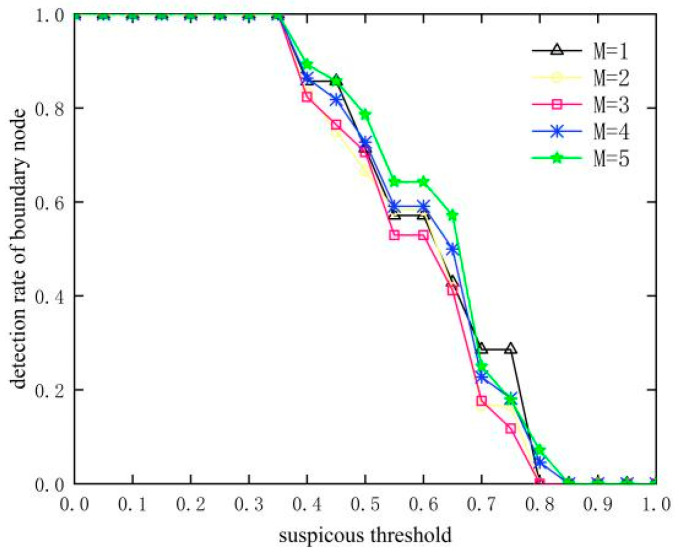
Influence of suspiciousness threshold on detection rate of boundary node.

**Figure 6 sensors-23-09709-f006:**
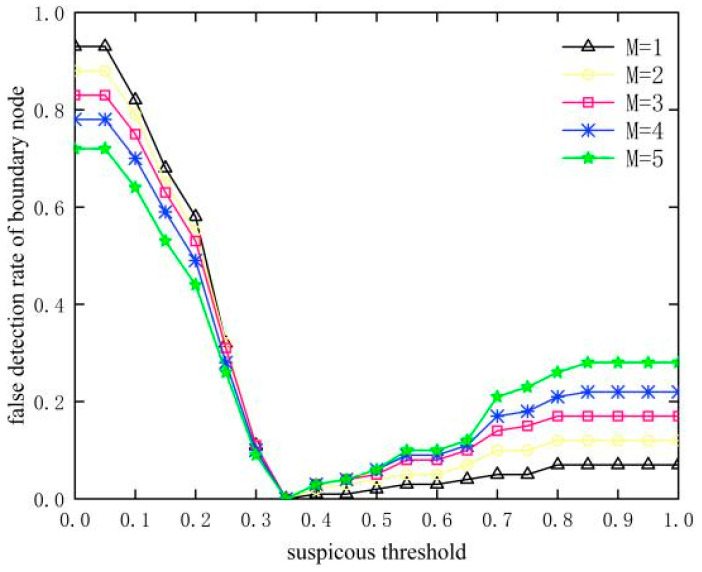
Influence of suspiciousness threshold on false detection rate of boundary node.

**Figure 7 sensors-23-09709-f007:**
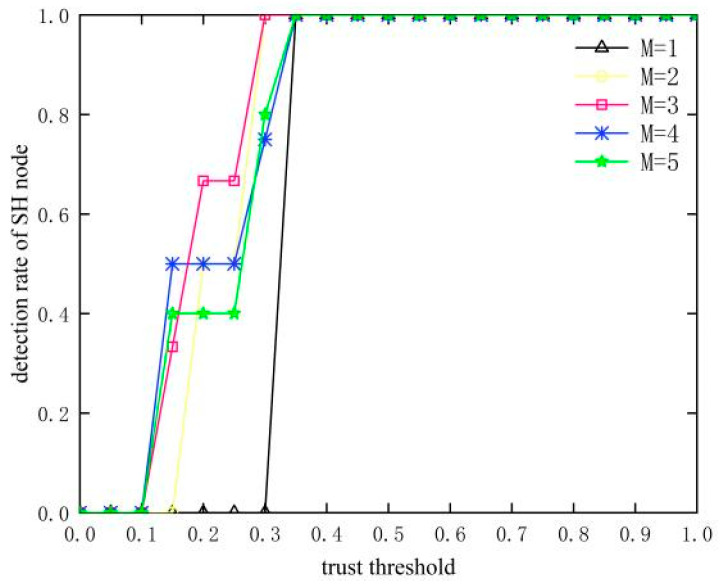
Influence of trust threshold on detection rate of SH nodes.

**Figure 8 sensors-23-09709-f008:**
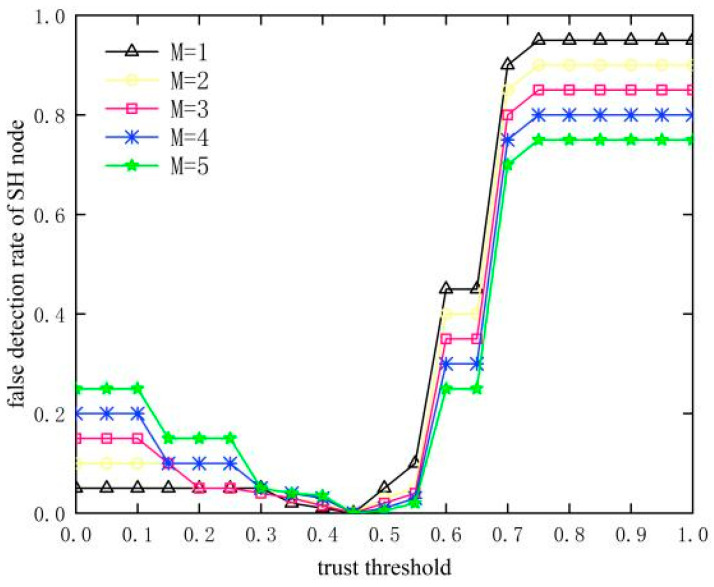
Influence of trust threshold on false detection rate of SH nodes.

**Figure 9 sensors-23-09709-f009:**
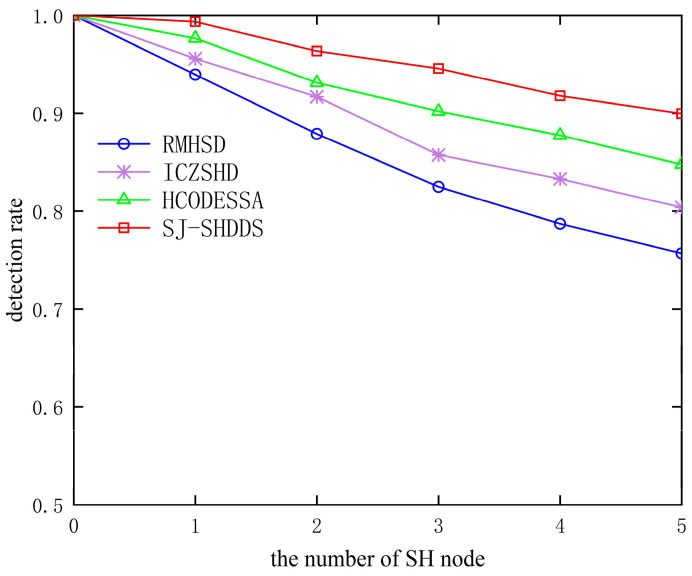
Comparison of detection rate of different algorithms.

**Figure 10 sensors-23-09709-f010:**
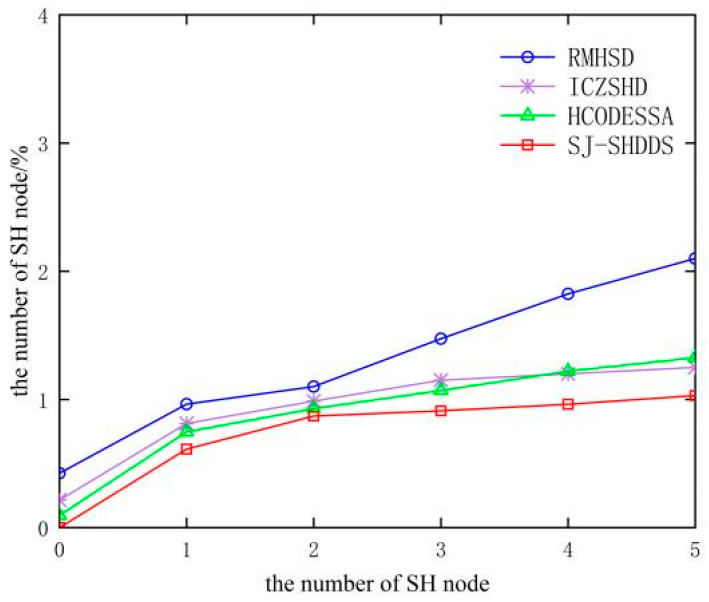
Comparison of false detection rate of different algorithms.

**Figure 11 sensors-23-09709-f011:**
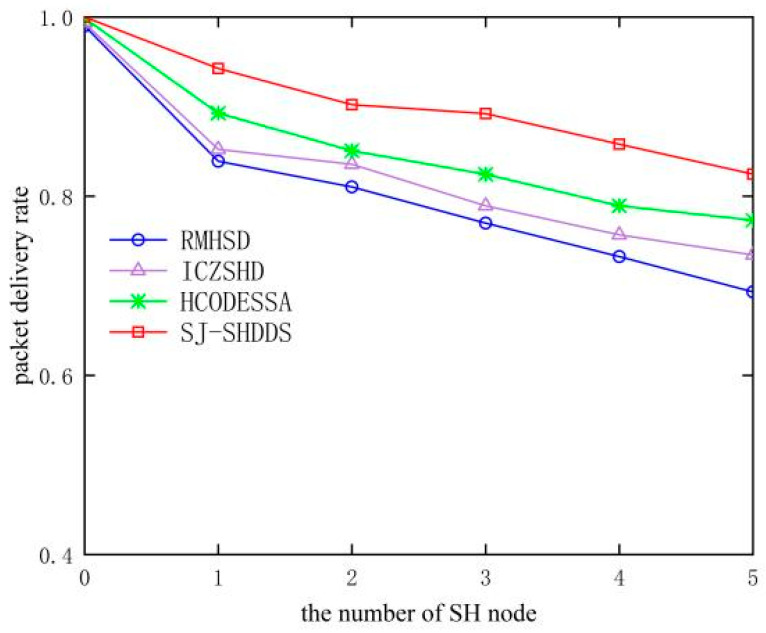
Comparison of packet delivery rate of different algorithms.

**Figure 12 sensors-23-09709-f012:**
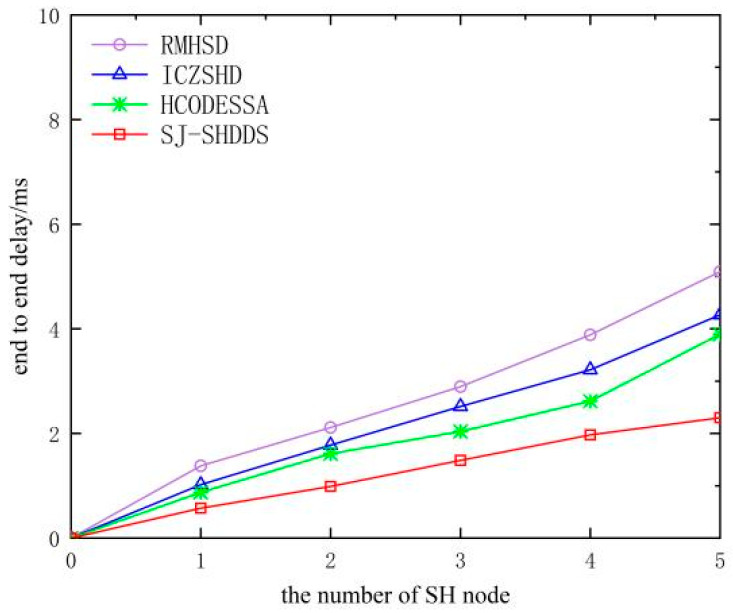
Comparison of end-to-end delay of different algorithms.

**Figure 13 sensors-23-09709-f013:**
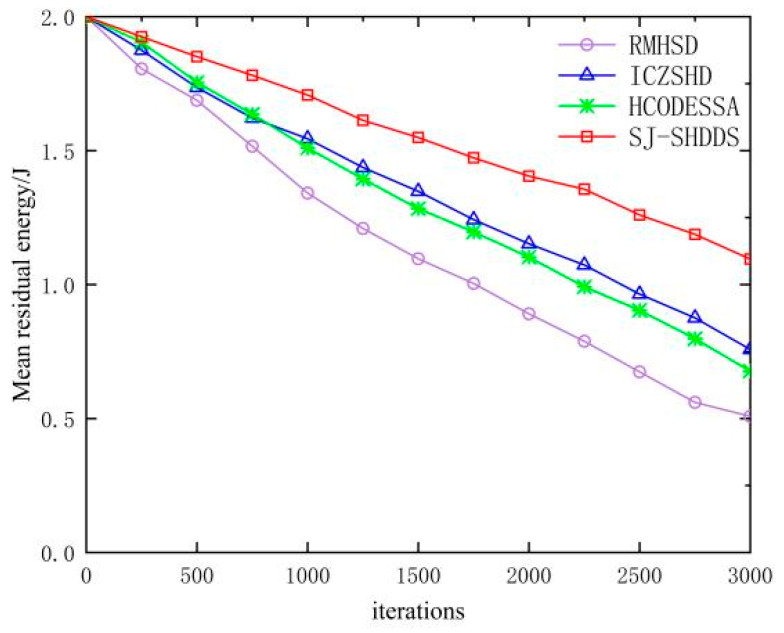
Mean residual energy.

**Table 1 sensors-23-09709-t001:** Information table for each node itself.

Node ID	Neighboring Node ID	Neighboring Node Hops	Suspiciousness
5	2	5	72%
	4	3	
	6	5	
	8	1	

**Table 2 sensors-23-09709-t002:** The relationship between trust level and indirect trust and connection value.

Trust Levels	Connectivity Interval	Connectivity Value	Indirect Trust
untrustworthy	A	[−1, −0.333]	0
uncertain	B	[−0.333, 0.333]	(Pessimistic potential) 0.25
			(Optimistic potential) 0.75
trustworthy	C	[0.333, 1]	1

**Table 3 sensors-23-09709-t003:** Simulation parameters.

Parameter	Numerical Value
Simulation area (m)	100 × 100
Total number of nodes	100
Number of SH nodes	1~5
Communication radius (m)	15
Initial energy (J)	2
Packet size (bit)	800
Transmission and reception energy consumption (nJ/bit)	50
Amplifier energy consumption (pJ/bit/m^2^)	10
*θ*	150

## Data Availability

Data are contained within the article.

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
