# Peer review of "Sinkhole Attack Defense Strategy Integrating SPA and Jaya Algorithms in Wireless Sensor Networks"

_sensors, 2023, doi:10.3390/s23249709_

Round 1

Reviewer 1 Report

Comments and Suggestions for Authors

Dear authors:

There are many points that came to my attention which I list the major ones below:

a) There is major a problem with citations. The authors tend to introduce equations without any citations, for example equations 2-6, The energy consumption of the sensor nodes. Please fix the citations of the equations through all the manuscript.

b) Figures need major improvement (quality).

c) Discuss possible limitations during your study, especially you are not using typical well known network simulator. 

d) Comparing with other algorithms HCODESSA, RMHSD, ICZHSD. I would appreciate if you explained figure 13 that discusses the mean residual energy. How did you obtain such results for , let say HCODESSA, given that they don't mention such quantity in their paper ( Reference 9). Same thing with figure 12 that deals with the end-to-end delay.

e) Please shed light on the results in Reference 8 (figure 12). Why your scheme deals with 5 sinkholes, while other schemes begin from 5 sinkhole to 30 as shown in reference 8. 

These are some points which I hope you can justify and fix in clear way. 

Comments on the Quality of English Language

I am more interested in the scientific content of the manuscript which I hope the authors will improve in a tangible way.

Reviewer 2 Report

Comments and Suggestions for Authors

Please see the file peer-review-32368851.v1.pdf below.

Comments on the Quality of English Language

There are grammar errors in English. Please review the entire paper to avoid similar errors. Below are a few examples (see highlighted section for errors)

(1)in the abstract:“Sinkhole attack is characterized by low difficulty to launch, high destructive power, and difficulty to detect and defend. It is a common attack mode for wireless sensor networks. This paper proposes a sinkhole attack detection and defense strategy integrating the SPA and Jaya algorithms in WSNs.”

(2) As the number of SH nodes M increases, more boundary nodes are affected, as shown in Fig 5.

(3)Line339, the nodes appearing as normal nodes being mistakenly detected as boundary nodes.”

Author Response

Thank you very much for your suggestion. Based on your suggestion, I have made modifications. Please take a look.

Round 2

Reviewer 1 Report

Comments and Suggestions for Authors

Thanks for the modifications.

Author Response

Thanks for your comments

Reviewer 2 Report

Comments and Suggestions for Authors

The author has made some modifications as requested, but I suggest that further modifications be made and the entire text be carefully reviewed before publication.  

(1)In last review, I suggest "In Figure 6, select a suspicious threshold of 0.35, and in Figure 8, select a trust threshold of 0.45. However, as the network size changes, these two values may vary. The author needs to add some data and figures to illustrate the impact of network size and node density on these two values."

     The authors did not answer my question. They should explain how the two values change with the network size. Because the actual network used is not always 100m * 100m.

(2)In last review, I suggested“When the Jaya algorithm first appears in the text, it should be explained where 

it comes from. But in the paper, after the word "Jaya" appeared several times, is cited on line 285.In 

addition, reference [21] provides an improved Jaya, and this article should cite the most original source 

of the algorithm.”

  The author response:"Thank you very much for raising this question. Although reference 21 is about the improved 

jaya algorithm, the original jaya algorithm has been briefly introduced in section 2.1. If you must trace 

back to its original jaya algorithm, perhaps you can find your answer in the following reference. Rao R. 

Jaya: A Simple and New Optimization Algorithm for Solving Constrained and Unconstrained 

Optimization Problems [J]. International Journal of Industrial Engineering Computations , 2016, 

7(1) :19-34."

      I suggest when introducing the Jaya algorithm in section 2.1, the original reference should be cited there. Indeed, I can search for the original reference about the Jaya, but when other readers have the same question, should they trace back to the original Jaya by themselves? 

(2) In lines 481-601, the context should be deleted. This error should not occur in the final version submitted.

(3)Table 3, lines 3,4,8,13, Wrapped text should be indented.

Comments on the Quality of English Language Line 23 :these information-->this information
